# U1RNP/lncRNA/Transcription Cycle Axis Promotes Tumorigenesis of Hepatocellular Carcinoma

**DOI:** 10.3390/diagnostics12051133

**Published:** 2022-05-03

**Authors:** Shun Li, Shuaiyin Zhang, Mingle Huang, Huanjing Hu, Yubin Xie

**Affiliations:** 1Department of Oncology, The First Affiliated Hospital, Sun Yat-sen University, No 58, Zhongshan Er Road, Guangzhou 510080, China; lish223@mail2.sysu.edu.cn (S.L.); huangml27@mail2.sysu.edu.cn (M.H.); 2Institute of Precision Medicine, The First Affiliated Hospital, Sun Yat-sen University, No 58, Zhongshan Er Road, Guangzhou 510080, China; zhangshy86@mail2.sysu.edu.cn (S.Z.); huhj6mail2@sysu.edu.cn (H.H.)

**Keywords:** hepatocellular carcinoma, long non-coding RNA, U1RNP, prognosis, transcription cycle

## Abstract

As a component of the spliceosome, U1 small nuclear ribonucleoproteins (U1RNPs) play critical roles in RNA splicing, and recent studies have shown that U1RNPs could recruit long non-coding RNAs (lncRNAs) to chromatin which are involved in cancer development. However, the interplay of U1 snRNP, lncRNAs and downstream genes and signaling pathways are insufficiently understood in hepatocellular carcinoma (HCC). The expression of U1RNPs was found to be significantly higher in tumors than normal tissues in liver hepatocellular carcinomas of The Cancer Genome Atlas (TCGA-LIHC) dataset. LncRNAs with potential U1-binding sites (termed U1-lncRNAs) were found to be mostly located in the nucleus and their expression was higher in tumor than in normal tissues Bioinformatic analysis indicated that U1-lncRNAs worked with RNA-binding proteins and regulated the transcription cycle in HCC. A U1-lncRNA risk model was constructed using a TCGA dataset, and the AUCs of this risk model to predict 1-, 3- and 5-year overall survival were 0.82, 0.84 and 0.8, respectively. Furthermore, silencing of the small nuclear ribonucleoprotein D2 polypeptide (SNRPD2) resulted in impaired proliferation, G1/M cell cycle arrest and downregulation of transcription-cycle-related genes in HCC cell lines. Taken together, these results indicate that U1RNPs interact with lncRNAs and promote the transcription cycle process in HCC, which suggests that these could be novel biomarkers in the clinical management of HCC.

## 1. Introduction

Hepatocellular carcinoma (HCC) is the second most fatal malignancy with a 5-year survival of 18% [1]. It was estimated that 905,677 cases were newly diagnosed as HCC and 830,180 patients died from HCC in 2020 [2]. Several risk factors, including infection by hepatitis virus, non-alcoholic fatty liver disease, excessive alcohol intake and tobacco use, have been widely acknowledged to contribute to the carcinogenesis of HCC [1,3]. With the rapid progress in molecular technology, the focus of HCC research has moved to studying the genetic and epigenetic alterations and to uncovering the transcriptome changes in the tumorigenesis of HCC. As a landmark cancer genomics program, The Cancer Genome Atlas (TCGA) has generated multiple omics data along with clinical data for over 30 types of human cancer and revealed a number of cancer driver genes that frequently are mutated or epigenetically altered in HCC [4].

In addition to genetic variants and epigenetic modifications, the aberrant expression of driver genes in HCC is generated by a complex network consisting of transcription factors, chromatin status, and non-coding RNA after the freshly synthesized messenger RNA (pre-mRNA) is spliced by the spliceosome, a complex that contains a large number of small nuclear ribonucleoproteins (snRNPs) [5]. Among the five different types of snRNPs, the U1 snRNP, which is more abundant than other snRNPs, is the first that binds to the pre-mRNA [6]. Interestingly, a recent study found that U1 homeostasis maintained the expression balance of normal genes and regulated the migration and invasion of cancer cells [7]. Another study found that the U1 snRNP promoted the recruitment of long non-coding RNAs (lncRNAs) to chromatin in a transcription-dependent manner [8].

Inspired by the novel functions of U1 snRNP recently uncovered by the above studies and the emerging evidence of lncRNAs in gene regulation in HCC [7,8,9], in this study we explored the interplay of U1 snRNP, lncRNAs and downstream genes by analyzing the TCGA dataset with the aim of better understanding the U1-related epigenetic network in the tumorigenesis of HCC. The biological functions of U1 snRNP that we discovered here were also confirmed in HCC cell lines.

## 2. Materials and Methods

### 2.1. Data Acquisition

The expression profiles, including RNA sequencing data, and the corresponding clinical data of HCC patients were downloaded from The Cancer Genome Atlas [4,10] (TCGA, https://portal.gdc.cancer.gov) (accessed on 1 June 2021) and the Gene Expression Omnibus (GEO) database [11] (dataset GSE76427, https://www.ncbi.nlm.nih.gov/geo) (accessed on 1 December 2021). The latter dataset was only used for the construction of a prognostic model.

### 2.2. Identification of U1-lncRNAs and Correlated mRNAs

We calculated the constitutive U1-binding site motif of the human genome in the 5′ splice site with 3nt in exons and 6nt in introns, according to [12]. We downloaded all annotated constitutive exons (exons without any known event of alternative splicing) of human genome version hg38 from the HEXEvent database [13] (http://hexevent.mmg.uci.edu/cgi-bin/HEXEvent/HEXEventWEB.cgi) (accessed on 1 October 2021). The sequence logo was plotted using the R packages ggplot2 and ggseqlogo.

Subsequently, we downloaded all the long non-coding RNA transcript sequences (version 38) from GENCODE (https://www.gencodegenes.org/) (accessed on 1 June 2021). We identified lncRNAs with U1-binding motifs by FIMO scanning [14] (https://meme-suite.org/meme/tools/fimo) (accessed on 1 October 2021). Then the sequence of all 18,090 lncRNAs from the above step were input in FIMO; we obtained 2749 lncRNAs with *p* values less than 0.0001.

We identified a total of 1027 eligible lncRNAs in the TCGA dataset which overlapped with the above list for subsequent analysis. Furthermore, we filtered lncRNAs and mRNAs according to the inclusion criterion of FPKM > 1 in more than 90% of samples in the TCGA-LIHC expression matrix. A total of 1017 lncRNAs and 7565 mRNAs were eligible in tumor tissues, while a total of 962 lncRNAs and 7535 mRNAs were eligible in normal tissues.

Afterwards, we calculated the Pearson’s correlation of the above lncRNAs and mRNAs for the expression matrix data of tumor (N = 371) and normal (N = 50) tissues, respectively. To identify tumor specific lncRNA-mRNA pairs, we set criteria of *p* < 0.05; |R| > 0.5. Eligible lncRNA-mRNA pairs in tumor tissues were included and then we excluded eligible lncRNA-mRNA pairs in normal tissues. We identified tumor specific lncRNA-mRNA pairs, for which a total of 327 lncRNAs were identified as tumor specific lncRNAs.

### 2.3. Cellular Location of lncRNAs with U1-Binding Motif

We further investigated the cellular location of the above lncRNAs with a U1-binding motif in the TCGA-LIHC dataset. Firstly, we examined the cellular location of all the 24,514 lncRNAs from the lncAtlas database [15] (https://lncatlas.crg.eu) (accessed on 1 October 2021) according to the mean cytoplasmic/nuclear value of all the cell lines tested. The number of nucleus-localized lncRNAs and cytoplasm-localized lncRNAs were 14,342 and 10,172, respectively. Then the intersect of the lncRNAs with the U1-binding motif and lncRNAs from lncAtlas was taken. A total of 509 lncRNAs with a U1-binding motif could be found in lncAtlas, with 382 lncRNAs nucleus-localized and 127 lncRNAs cytoplasm-localized. A hypergeometric test was performed using the function HYPGEOMDIST in Excel to compare the distribution of lncRNAs with a U1-binding motif and all lncRNAs from lncAtlas. Finally, we identified a total of 175 nuclear U1-related tumor specific lncRNAs from the 327 tumor-specific lncRNAs and termed these U1-lncRNAs for subsequent analysis.

### 2.4. Functional Annotation and Regulation Network Construction

Gene Ontology (GO), Metascape (https://metascape.org) (accessed on 1 December 2021) and ClueGo [16] were used for the functional analysis of mRNAs. The GO annotation was performed by the R package clusterProfiler based on genes from the “GO-BP, MF and CC” ontology. The Metascape analysis was performed online and ClueGo was performed in Cytoscape. The interaction data between lncRNA and target RNA was taken from the Encori database [17] (https://starbase.sysu.edu.cn) (accessed on 1 June 2021). The RNA binding protein list was downloaded from RBPTD [18] (http://rbptd.com/#/) (accessed on 1 December 2021). The GO level count was performed by TBtools [19] based on data downloaded from http://geneontology.org/docs/download-ontology/ (accessed on 1 December 2021); only Level 5 GO terms in the GO-MF ontology category were counted. The clip-seq interaction data between RNA-binding proteins and target RNA was from the Encori, NPInter [20] (http://bigdata.ibp.ac.cn/npinter4/) (accessed on 1 June 2021) and POSTAR [21] (http://111.198.139.65/) (accessed on 1 June 2021) databases.

The list of selected genes in the transcription cycle pathway were from a reference study [22] which contained a total of 108 related genes (a detailed list can be found in Appendix A). The enrichment scores of the transcription cycle of tumor and normal samples in TCGA-LIHC were calculated using the R packages GSVA and GSEABase. The networks were displayed in Cytoscape [23].

### 2.5. Construction of the Prognostic Based on U1-lncRNAs

The U1-lncRNAs and associated mRNAs shared by both TCGA and GSE76427 [11] datasets were selected as input to the LASSO regression model. The R package glmnet was used to determine the corresponding coefficients for different U1-lncRNAs in the model. A risk score of the prediction model was used to divide patients in each cohort into high or low risk groups. The predictive efficiency of the model was assessed by survival curve, risk curve and receiver operating characteristic (ROC) curve analysis through the R packages Survival and Survival ROC. The R package forestplot was used to display the results of the univariate and multivariate Cox regressions.

### 2.6. Cell Culture and Transfection

The hepatocellular carcinoma cell lines HuH-7 and SUN-449 were from ATCC. HuH-7 cells were grown in DMEM media with 10% fetal bovine serum, 100 U/mL penicillin, and 100 μg/mL streptomycin. SUN-449 cells were grown in RPMI 1640 with 10% fetal bovine serum, 100 U/mL penicillin, and 100 μg/mL streptomycin. These cells were cultured at 37 °C in a 5% CO_2_ environment. At a time 12 h before transfection, 2 × 10^5^ cells per well were placed into six-well plates. According to the manufacturer’s instructions, we employed Lipofectamine 3000 to transfect the small interfering RNA (siRNA) sequences. The following siRNA sequences (Suzhou GenePharma, Suzhou, China) were used: siRNA for the negative control (NC) (5′-UUCUCCGAACGUGUCACGUTT-3′) and siRNA for SNRPD2 (5′-GCGAGAGGAGGAGGAAUUUTT-3′).

### 2.7. Cell Proliferative Assay

Cells transfected with SNRPD2 siRNA or siNC were seeded in a 96-well plate at 2000 cells per well and incubated. A quantity of 10 μL of Cell Counting Kit-8 solution (Yeasen Biotechnology, Shanghai, China) was mixed with fresh media and added to each well and incubated for 2 h. A microplate reader (VarioskanTM LUX, Thermofisher, Waltham, MA, USA) was used to measure the absorbance at 450 nm every day. Each measurement was performed in five replicates.

### 2.8. Cell Cycle Analysis

Cells were cultured in six-well culture with 2 × 10^5^/well cells for 12 h and were transfected with SNRPD2 siRNA or siNC. The cells were collected using trypsin after 48 h, washed twice with phosphate-buffered saline, and then fixed into 70% pre-cooled ethanol at 4 °C for 12 h, stained with RNase A-containing PI buffer (Yeasen Biotechnology, Shanghai, China) for 30 min, and analyzed using flow cytometry (Beckman Coulter Quanta SC System, Indianapolis, IN, USA). The results were analyzed using FlowJo software (flowjo version10.6.2, Ashland, OR, USA).

### 2.9. Quantitative Real-Time Polymerase Chain Reaction

Total RNA was extracted using TRIzol reagent (Invitrogen Corporation, Waltham, CA, USA) 48 h after siRNA transfection, and then reverse-transcribed into cDNA using PrimeScriptRT reverse transcriptase (Takara, Shiga, Japan). The cDNA templates were diluted with RNase-free water for (1:3) and combined with SYBR Green premix with RoxII (Takara, Shiga, Japan) to perform quantitative-PCR reactions according to the manufacture’s protocol and measured by the Applied Biosystems™ QuantStudio™ 5 system (Thermofisher, Waltham, MA, USA). The expression of GAPDH was used as the endogenous control for the mRNA levels. The relative expression levels were calculated using the 2^−^^△△Ct^ methods. The primers for qPCR were as follows: SNRPD2-Forward 5′-CAAGTGCTCATCAACTGCCGCA-3′ and SNRPD2-Reverse 5′-GCGGTCTTTGTTGACTGGCTTG-3′; GAPDH-Forward 5′-ACAACTTTGGTATCGTGGAAGG-3′ and GAPDH-Reverse 5′-GCCATCACGCCACAGTTTC-3′; CDK1-Forward 5′-GGAAACCAGGAAGCCTAGCATC-3′ and CDK1-Reverse 5′-GGATGATTCAGTGCCATTTTGCC-3′; CCNH-Forward 5′-CGATGTCATTCTGCTGAGCTTGC-3′ and CCNH-Reverse 5′-TCTACCAGGTCGTCATCAGTCC-3′; TAF1-Forward 5′-GGCTAAAGCTCTGCGCTGACTT-3′ and TAF1-Reverse 5′-AGCACTGCTCTGGTGACACCAT-3′.

### 2.10. Western Blot

Whole cell lysis was collected by pre-cooled RIPA with a protease inhibitor and a phosphatase inhibitor. The protein samples were separated by sodium dodecyl sulfate polyacrylamide gel electrophoresis (SDS-PAGE) and then transferred onto polyvinylidene fluoride (PVDF) membranes. Next, the membranes were washed with Tris-buffered saline/Tween 20 (TBST) three times and blocked with 5% BSA dissolved in TBST at room temperature for one hour. Then, the membranes were incubated with Anti-SNRPD2 (1:3000 diluted; Abcam #ab198296, Cambridge, UK) or anti-β-Tubulin (1:5000 diluted; CST #2146S, Danvers, MA, USA) overnight at 4 °C. After washing three times with TBST for 5 min each, horseradish peroxidase (HRP)-conjugated secondary antibodies (1:10,000 diluted; Abcam #ab205718, Cambridge, CB2 0AX, UK) were added for 1 h at room temperature. After three washes with TBST, the protein expression levels were evaluated with a chemiluminescence reagent using Amersham™ ImageQuant™ 800 (General Electric Company, Boston, MA, USA). The grey scale of each lane was measured by ImageJ and a T-test was used to compare the statistical significance.

### 2.11. Statistical Analysis

R (version 4.1.0) and SPSS 26.0 (IBM SPSS Statistics for Windows, version 26.0 (IBMCorp., Armonk, NY, USA) were used to conduct all statistical analyses. For the boxplot of the expression of U1RNPs in tumor and normal tissues, we normalized the FPKM of each gene by log2(FPKM + 1) and a T-test was used for comparison. For comparison of the overlap genes between U1-lncRNA-associated mRNAs and the RBPTD gene list, Fisher’s exact test was used; the background gene number was set as 19,969 [24]. For comparison of expression level and enrichment score, the R packages ggplot2 and ggpubr were used to display the results and the R package ggsignif was used for calculation and annotation of statistical significance. For correlation analysis, the Pearson test was used. Least absolute shrinkage and selection operator (LASSO) regression were used to calculate the prognostic signature’s coefficients. The HRs were calculated using univariate Cox proportional hazards regression. The Kaplan–Meier technique was used to create the survival curve. The log-rank test was used to assess overall survival (OS) differences. A *p*-value of 0.05 was used to indicate a significant difference in the statistical analysis, with the confidence interval (CI) set at 95%. All cell experiments were conducted for three biological replicates.

## 3. Results

### 3.1. Overview of U1RNPs and Its Related lncRNAs in HCC

The design of the present study was shown in Figure 1A. Briefly, we identified U1-related tumor-specific lncRNAs in the TCGA-LIHC dataset with their correlated mRNAs. Then we analyzed the expression and location of U1-related tumor-specific lncRNAs and performed a functional annotation of the correlated mRNAs. Additionally, regulation networks of U1-related tumor specific lncRNAs and their correlated mRNAs were constructed. We constructed a prognosis prediction model based on the shared lncRNAs of TCGA and the GEO dataset. The expression of 10 core component U1RNPs (SNRPA, SNRPB, SNRPC, SNRPD1, SNRPD2, SNRPD3, SNRPE, SNRPF, SNRPG and SNRNP70) were significantly higher in tumor tissues than normal tissues (Figure 1B). The U1-binding motif was calculated and displayed in Figure 1C. A total of 1027 lncRNAs were found to possess a U1-binding motif by FIMO scanning, and 327 among these were determined to be tumor-specific. Among the U1-related tumor-specific lncRNAs, 36 lncRNAs were found to be significantly correlated with U1RNPs (*p* < 0.05; |R| > 0.5) in the TCGA-LIHC dataset (Figure 1D). These results implied a close interaction between the U1RNPs and lncRNAs harboring a U1 binding motif.

### 3.2. Distribution of HCC-Specific lncRNAs with U1 Binding Sites

Given that U1RNP regulates the chromatin retention of lncRNAs, indicating the nuclear localization and function of these lncRNAs, the lncRNAs with a U1 binding motif were found to be predominantly (382/509, 75.05%) located in the nucleus, according to the LncAtlas Database (Figure 2A). A hypergeometric test was performed to compare the distribution of lncRNAs with a U1-binding motif and all lncRNAs from lncAtlas. The statistical *p*-value for the hypergeometric test was less than 0.0001, which implies a significantly higher nucleus-localized ratio of the lncRNAs with a U1 binding motif. In addition, those lncRNAs that were significantly correlated with U1RNPs were mainly nucleus-localized (Figure 2B). Thus, we collectively compared the expression level of lncRNAs with a U1 binding motif between tumor and normal tissues in the TCGA-LIHC dataset. The expression of these lncRNAs was higher in tumor than in normal tissues (Figure 2C). We focused on these nuclear U1-related tumor specific lncRNAs (U1-lncRNAs, hereafter) and examined the correlation between U1-lncRNAs and mRNA. Filtering with a Pearson’s correlation coefficient of more than 0.5 or less than −0.5, a total of 3453 mRNAs were found to be significantly correlated with U1-lncRNAs (3252 mRNAs were positive correlated and 201 mRNAs were negative correlated).

### 3.3. Function and Network Analysis of U1-lncRNAs and Correlated mRNAs

To further investigate the function of correlated mRNAs of the U1-lncRNAs, GO functional annotation was implemented in the above positively and negatively correlated mRNAs, respectively. Genes in the positive set were involved in RNA splicing, mRNA catabolic process, focal adhesion and histone binding (Figure 3A), while those in the negative set were mainly related to small catabolic processes and cellular respiration (Figure 3B). We also attempted to unearth the cis- and trans-regulation network of U1-lncRNAs and their correlated mRNAs (Appendix A). For cis-regulation of U1-lncRNAs, we examined the genomic distance between U1 lncRNAs and mRNAs in less than 10 kb [25] and found 28 pairs of cis-acting lncRNAs and mRNAs. For trans-regulation of U1-lncRNAs, we searched target mRNAs of U1-lncRNAs in the Encori database and found 82 pairs of lncRNA-mRNA in all. A Metascape functional analysis was performed for mRNAs in the cis- and trans-pairs. The genes regulated by trans-acting U1-lncRNAs were involved in mitotic cell cycle phase transition, TGF-beta signaling, and transcription initiation (Appendix A), while genes regulated by cis-acting U1-lncRNAs were involved in histone modification and protein glycosylation (Appendix A).

### 3.4. U1-lncRNAs Activates the Transcription Cycle to Promote HCC

According to the above enrichment results, altered pathways involved in the regulation of U1-LncRNAs were principally related to RNA-processing. We counted GO terms and confirmed that RNA-binding was the most frequent term (Figure 4A), indicating the intimate interaction between U1-LncRNAs and RNA-binding processes. Furthermore, among the correlated mRNAs of U1-lncRNAs, 851 were found in the RBPTD, accounting for 43% (851/1999) of the RBPs identified in the RBPTD (Figure 4B); the overlap was significant (Fisher’s exact test, *p* < 0.001). To investigate the interaction of these RBPs and their target genes in the downstream mRNA network of U1-LncRNAs, the RBP-target interactions with CLIP-seq level clue were validated in Encori, NPinter, and POSTAR databases. A total of 100 RBPs and 7000 target mRNAs within 590,000 interactions were confirmed. Subsequently, the correlation of the above RBP-mRNA pairs was calculated focusing on the top 500 pairs after filtering by significance (*p* < 0.05 and *p*.adj < 0.05) (Figure 4C). To investigate the function of the target genes in the RBP-mRNA pairs, a ClueGo analysis was performed by Cytoscape (Figure 4D). It revealed rRNA-binding, mRNA-binding and transcription factor binding, indicating the essential role of the activated transcription cycle in HCC tumors. To confirm the above results, a set of genes involved in the transcription cycle was determined and then the enrichment score of the transcription cycle was calculated in each sample in the TCGA-LIHC datasets. The scores were significantly higher in the tumor tissues than in the normal tissues (Figure 4E), demonstrating that U1-LncRNAs could facilitate HCC tumorigenesis by activating the transcription cycle.

### 3.5. Construction and Validation of the U1-lncRNAs-Based Prognostic Model

To evaluate the prognostic value of U1-lncRNAs and associated mRNAs, we applied the LASSO (least absolute shrinkage and selection operator) Cox regression model to construct a prognosis prediction model according to the expression level of lncRNAs. In addition, shared lncRNAs and mRNAs by the TCGA dataset and GSE76427 dataset were determined before the LASSO selection (Appendix A). A total of 24 mRNAs and lncRNA in the TCGA dataset were selected into the U1-LncRNA risk model by LASSO (Appendix A) with sixteen found to be protective factors and eight to be risk factors, respectively (Figure 5A). Subsequently, patients were divided into a high-risk group or a low-risk group according to the LASSO-COX risk score. Afterwards, the U1-lncRNA risk was validated to be an independent predictor of OS (hazard ratio 3.81 [95% CI 2.56,5.68]) in a Cox proportional hazards model (Figure 5B). Compared with patients with high-risk scores, those with low-risk scores demonstrated significantly more favorable prognosis in the TCGA cohort (Figure 5C). Similarly, the U1-LncRNA risk model performed well in its capacity to predict overall survival with the 1-, 3- and 5-year AUCs found to be 0.82, 0.84 and 0.8, respectively (Figure 5D). As for the GSE76427 external validation cohort, patients were categorized using the same U1-LncRNA risk model. The difference in the overall 5-year survival time between the high- and low-risk groups in the GSE76427 cohort did not reach significance (*p* = 0.2) (Appendix A). The 1-, 3- and 5-year AUCs of the U1-lncRNA risk model in the GSE76427 cohort were 0.71, 0.71 and 0.62, respectively (Appendix A).

### 3.6. Biological Function of SNRPD2 in HCC Cells

To investigate the contribution of U1RNP in the development of HCC in vitro, SNRPD2 knockdown was performed by siRNA in two hepatocellular carcinoma cell lines (HuH-7 and SNU-449). Firstly, the knockdown efficiency of siRNA targeting the SNRPD2 was determined by qRT-PCR (Figure 6A) and Western blot analysis in HuH-7 and SNU-449 (Figure 6B), respectively. Next, we examined the proliferation by CCK8 assay and found that the cell growth was significantly inhibited in the siRNA group compared with the NC group (Figure 6C). Similarly, siRNA resulted in significant cell cycle arrest in the G1/M phase compared with the NC group detected by flow cytometry assay (Figure 6D). We also checked whether genes in the transcription cycle altered after SNRPD2 knockdown by qRT-PCR. The expression levels of CDK1, CCNH and TAF1 were significantly elevated in the siRNA group compared with the NC group (Figure 6E). These results demonstrated that the overall transcription cycle of the HCC cells was impaired by the attenuated U1RNP.

## 4. Discussion

Whereas both U1RNPs and lncRNAs play essential roles in the development of HCC, little is known about the interaction of U1RNPs and lncRNAs or their contribution to HCC carcinogenesis. Our results have uncovered that specific U1-lncRNAs, whose chromatin retention was maintained by U1RNPs, could orchestrate and activate the transcription cycle, resulting in HCC tumorigenesis. Based on the substantial samples in the TCGA-LIHC datasets, the intimate communication between U1RNPs and specific lncRNAs was supported by our bioinformatic analysis. These U1-lncRNAs could mainly function as regulators of RBP and related genes to boost the transcription cycle. A prognosis prediction model including several U1-lncRNAs was able to discriminate the overall survival time of patients with high or low risk scores, which was validated by an independent external GSE dataset. Furthermore, the activated transcription cycle in the HCC cells was demonstrated to be inhibited by silencing SNRPD2 in vitro, which is one component of the U1RNPs.

It has been found that the abundance of U1RNPs dramatically exceeds other major factors of the spliceosome, such as U2, U3, U5 and U6 RNPs, which suggests an extra role in addition to the initial recognition of pre-mRNAs [6,8]. An increasing number of studies have revealed that higher expression of components of U1RNP was associated with worse prognosis in multiple cancers [26,27,28,29,30,31,32,33]. These studies indicated the canonical function of U1RNPs in the regulation of splicing. Furthermore, numerous lncRNAs have been identified to play essential roles in the development of HCC and other cancers [34,35,36,37,38]. While the mechanism of how U1RNPs can modulate the chromatin retention of lncRNAs has been elucidated [8], this mechanism has barely been studied in cancer. Here, by scanning the U1-binding motif in the sequences of lncRNAs, eligible lncRNAs were found to predominantly localize in the nucleus. Moreover, the expression of these lncRNAs was significantly correlated with U1RNPs. The above two findings suggest that U1RNPs tether lncRNAs and help them to carry out functions in the nucleus.

With respect to the interaction of U1-lncRNAs and mRNAs, firstly, we found that U1-lncRNAs positively correlated with genes which were involved in mRNA catabolic pathways and the cell cycle. In addition, nearly one fifth of these genes were responsible for RNA-binding. These results indicated that the abnormal regulation of the core transcriptional machinery was induced by U1-lncRNAs in part, which has recently been termed transcription cycle regulation [22]. We observed significantly more elevated enrichment scores in HCC tumor tissues than in normal tissues. In addition, the expression of several components in the transcription cycle, including CDK1, CCNH, and TAF1, was able to be impaired by SNRPD2 silencing. Therefore, U1RNPs and related lncRNAs facilitating the development of HCC by boosting the transcription cycle are potential therapeutic targets for selective transcription perturbation. Except for inhibitors of transcriptional cyclin-dependent kinases and the mediator complex [22], the U1RNP components responsible for the initial step of splicing could be potential targets [39]. Yet the current targets of the splicing process are mainly splicing factors and kinases, in connection with which several clinical trials have been conducted [40,41]. According to our study, bromodomain and extra-terminal inhibitors (BETis) may be promising drugs for HCC patients due to their potential effects on the modulation of splicing, histone modification and transcription elongation [42]. Moreover, U1-lncRNAs could be inhibited by antisense oligonucleotides (ASOs) [43,44,45] with recent progress in nanoparticle-based delivery methods [46].

Intriguingly, U1RNPs were reported to interact with energy metabolism in cellular biology [47]. In our analysis, the U1-lncRNA negatively correlated genes were consistently related to the oxidative phosphorylation pathways. Our findings also supported the critical role of U1RNPs with U1-lncRNAs in energy metabolism, given that the shift from OXPHOS to glycolysis is the hallmark of cancer cells.

Although our analysis was based on a TCGA dataset with an appreciable sample size and was validated in multiple databases, there are several limitations to our study. The first is that the RNA-Seq data of the TCGA LIHC project was not strand-specific, which hampers the quantitative determination of lncRNAs to some extent, especially anti-sense lncRNAs. In addition, while the prognostic value of U1-lncRNAs and associated mRNAs was demonstrated, the difference in overall 5-year survival time between high and low risk group in GSE76427 cohort did not reach significance (*p* = 0.2). A possible explanation for this might be the different methods of measurement and normalization of the TCGA cohort and the GSE76427 cohort (RNA-Seq vs. MicroArray). Elevated performance of the model is expected in more compatible datasets with more shared lncRNAs.

In summary, our study is the first to demonstrate that U1RNP-related lncRNAs induce the aberrant transcription of HCC, providing additional potential targets for precision therapy for patients with HCC. In addition, the prognostic valve of these lncRNAs has been confirmed, which may contribute to further prediction of the prognosis of HCC patients.

## Figures and Tables

**Figure 1 diagnostics-12-01133-f001:**
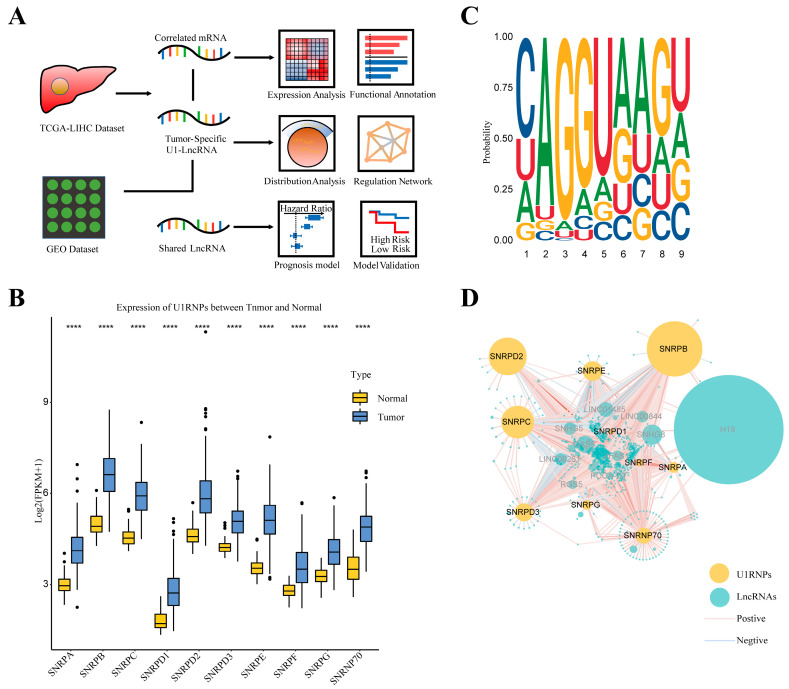
Overview of the U1RNPs and U1-lncRNAs in HCC. (**A**) Flowchart of the study. (**B**) Box plot of the expression level of U1RNPs. The expression of U1RNPs in each sample was normalized by log2(FPKM + 1) and T-test was used for comparison. Significance level: **** *p* < 0.0001. (**C**) Sequence logo of the U1-binding motif. (**D**) The correlation network of U1RNPs and U1-lncRNAs (The color and width of lines were mapped with the correlation and the size of pies was mapped with the expression level).

**Figure 2 diagnostics-12-01133-f002:**
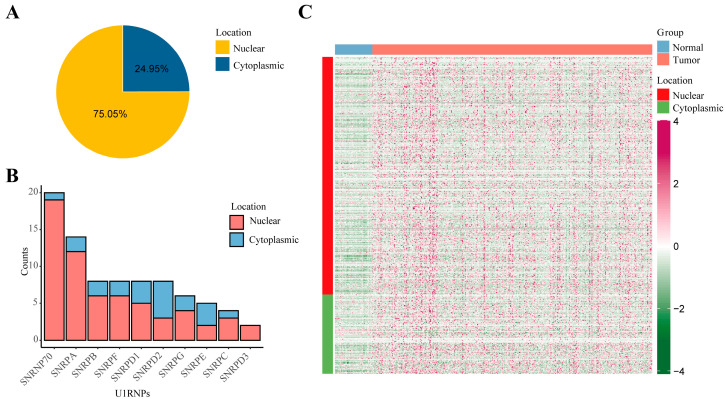
Distribution and expression of U1-lncRNAs in HCC. (**A**) Pie plot showing location of tumor specific lncRNAs with U1-binding motif in LncAtlas database. (**B**) Bar plot showing the overall significant counts and location of correlated lncRNAs of each U1RNP based on the correlation of their expression level. The criteria for significant correlation were: *p* < 0.05; |R| > 0.5. (**C**) Heatmap showing the expression and location of tumor specific lncRNAs within tumor and normal samples.

**Figure 3 diagnostics-12-01133-f003:**
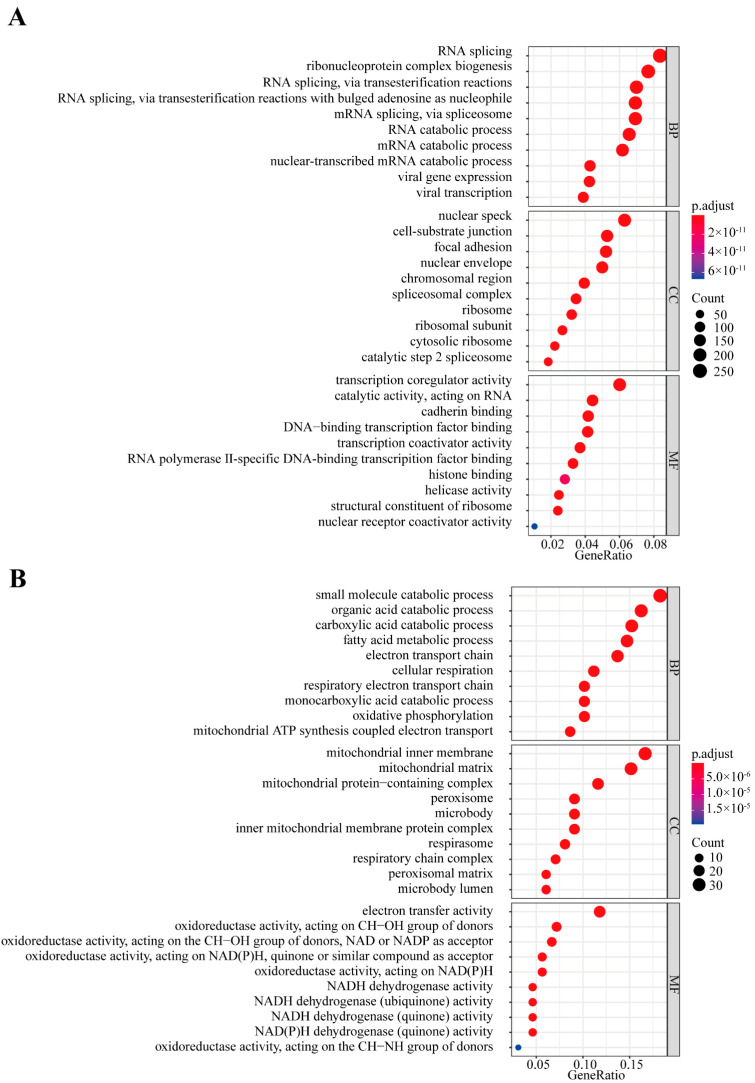
Functional annotation and lncRNA-mRNA interaction network. (**A**) GO annotation of mRNAs that were positively correlated with U1-lncRNAs. (**B**) GO annotation of mRNAs that were negatively correlated with U1-lncRNAs.

**Figure 4 diagnostics-12-01133-f004:**
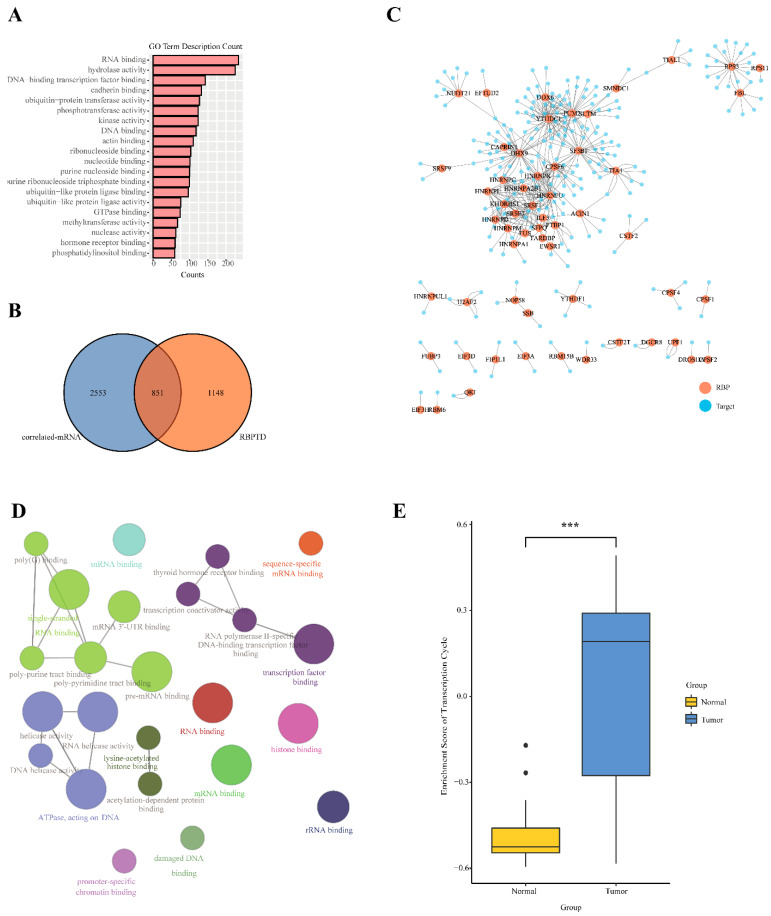
The RBP-regulated network and transcription cycle pathway of U1-lncRNAs correlated mRNAs. (**A**) GO term count of U1-lncRNAs correlated mRNAs. (**B**) Venn diagram illustrating the overlapped genes in U1-lncRNAs correlated mRNAs and RBPTD database. (**C**) RBP-target network constructed by the top 500 pairs of RBP-target in U1-lncRNAs correlated mRNAs. (**D**) Functional annotation of genes in the top 500 pairs by ClueGO. (**E**) The enrichment score of transcription cycle of tumor and normal samples in TCGA-LIHC dataset. T-test was used to compare enrichment scores of tumor and normal samples. Significance level: *** *p* < 0.001.

**Figure 5 diagnostics-12-01133-f005:**
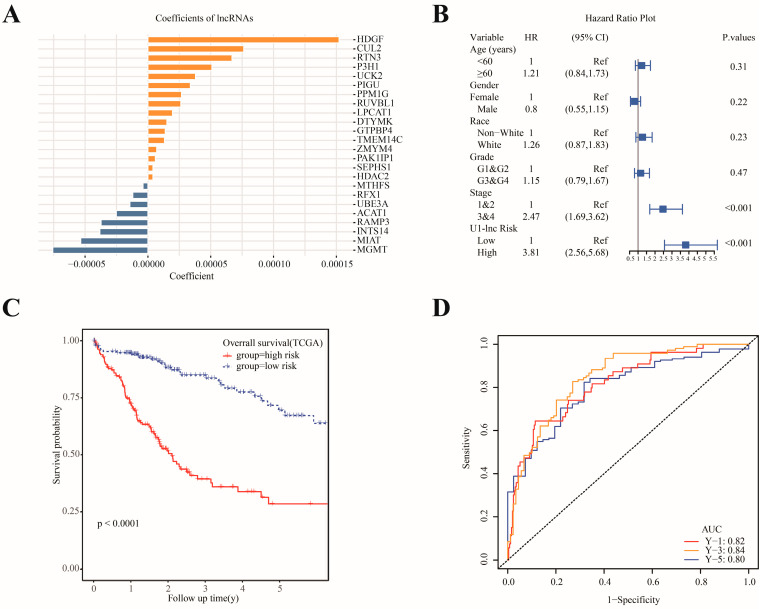
Construction of a prognostic model based on U1-lncRNAs and associated mRNAs to predict the survival of patients in TCGA-LIHC cohort. (**A**) Coefficients of selected U1-lncRNA and associated mRNAs by LASSO. (**B**) Univariate Cox regression analysis of OS for the U1-lncRNA risk score in addition to clinical features. (**C**) Kaplan–Meier curves of OS between the high-risk and low-risk groups in the TCGA cohort. (**D**) ROC curve of 1-, 3- and 5-year survival of the risk model in TCGA cohort.

**Figure 6 diagnostics-12-01133-f006:**
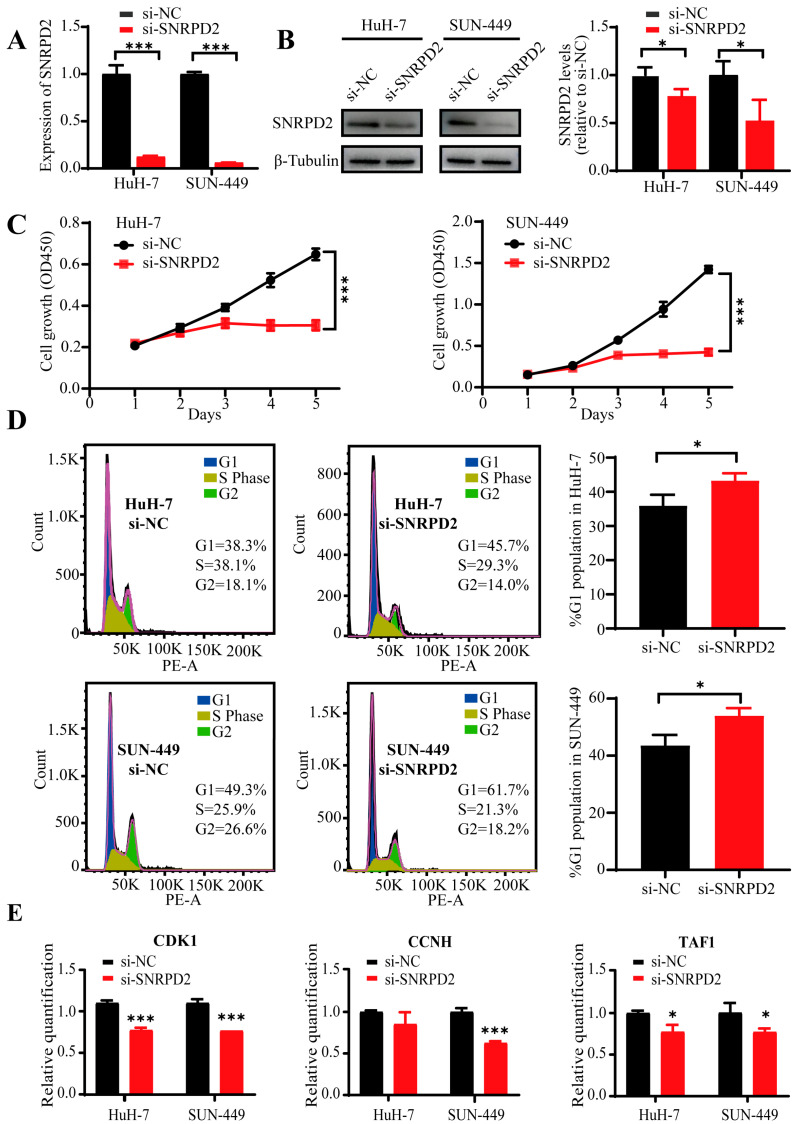
Exploration of the function of SNRPD2 in HCC cell lines. (**A**) SNRPD2 knockdown (KD) efficiency of siRNA detected by qRT-PCR. (**B**) SNRPD2 knockdown efficiency of siRNA detected by Western blot; the statistical result of quantification is shown on the right. (**C**) Proliferation of SNRPD2 KD and NC group detected by CCK8 in HuH-7 and SNU449, respectively. (**D**) Cell cycle of SNRPD2 KD and NC group detected by flow cytometry in HuH-7 and SNU449, respectively. Ratio counts of G1 phase are displayed on the right. (**E**) Detection of transcription cycle related genes after SNRPD2 KD by qRT-PCR in HuH-7 and SNU449, respectively. T-test was used for comparison. Significance level: * *p* < 0.05, *** *p* < 0.001.

## Data Availability

The datasets [TCGA-LIHC and GSE76427] for this study can be found in the GDC Data Portal [https://portal.gdc.cancer.gov/] (accessed on 1 June 2021) and Gene Expression Omnibus [https://www.ncbi.nlm.nih.gov/geo/] (accessed on 1 December 2021).

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
