# Peer review of "U1RNP/lncRNA/Transcription Cycle Axis Promotes Tumorigenesis of Hepatocellular Carcinoma"

_diagnostics, 2022, doi:10.3390/diagnostics12051133_

Round 1

Reviewer 1 Report

In this manuscript, the authors presented bioinformatic analyses to explore the relationship between lncRNAs and U1RNPs in hepatocellular carcinoma. They found that lncRNAs associated with U1RNPs tend to have higher expression in tumors, and they cooperate with RNA-binding proteins to regulate the transcription cycle. They also built a risk model with these lncRNAs to predict overall survival, and further validated the impairment of the transcription cycle after knockdown of one of the U1RNPs, SNRPD2. These results are of interest to readers in the field. However, the methods of these analyses were not described clearly, which are critical issues that need to be addressed before publication.

Major issues:

  1. The concept of U1-lncRNAs is the key to all the analyses in the manuscript, but the authors did not give adequate descriptions of how they defined U1-lncRNAs. Specifically:
    1. In line 64, how exactly was the U1-binding motif computed? What are the "constitutive exons" used to compute the 9-nt motif? The HEXEvent is a database of exon splicing events published in 2013 and the website seems to be down now. I can only guess the authors used the sequences at splicing events and checked for motif enrichment. But without further details, there is no way to tell.
    2. In lines 66-68, what were the parameters used in FIMO? These will determine how stringent it was when calling the U1 motif is "in" the lncRNAs. Also, how did the authors define lncRNAs and how many lncRNA were used? Did they use annotations from GENCODE or did they manually curate the list (by length for example)?
    3. In lines 69-71, it seems that the Pearson's correlations were computed for each lncRNA-mRNA pair, across samples. How many tumor and normal samples in the TCGA dataset were used to compute the correlations? What are the standards to call the correlations to be significant? It seems they used p < 0.05 and |R|>0.5 (stated in main text line 177). Are those p-values adjusted for multiple comparisons (it seems to be raw p-values)? These are extremely important because the author said "If the correlation of lncRNA-mRNA pair was significant only in tumor but not in normal tissue, the lncRNA was identified as tumor specific lncRNA." Note that from Fig.2C it looks like there are much more tumor samples compared to normal samples. This will bias the correlation test where the expression of the to-test lncRNA-mRNA pair in the tumor samples might be easier to reach a significant correlation even the R values are relatively low. Also, it's unclear whether both tumor and normal samples or only tumor samples were used to compute Pearson correlation for the subsequent analyses, once the tumor-specific lncRNAs were defined.
  2. In Fig.1B, how was the differential expressed tested? Did the authors run a standard RNA-seq DE test (like edgeR, DEseq2, etc.) or it was just a simple t-test or Wilcoxon test with the FPKM? How was "significantly higher" defined (in line 173)? What are the meaning of those asterisks on top (FDR<0.001?) These should be stated clearly in the methods and the figure legend.
  3. In Fig.2A and line 189, the authors said the U1-lncRNAs are predominantly located in the nuclear. What are the distributions of the nuclear/cytoplasm localization for other lncRNAs in the LncAtlas database? Without some sort of proper background to compare with, it's hard to see whether the preference in the nuclear is a property of the U1-lncRNAs or it's universal for all lncRNAs.
  4. In the figure legend of Fig.2B, what does "according to their expression level" mean? It seems that the counts in the bar plots are the numbers of lncRNAs associated with each of those U1RNPs. No gene expression levels (either the lncRNAs or the U1RNPs) were shown here.
  5. What were the background genes to test against in the GO enrichment analysis? 
  6. In Fig.4A and lines 233-234, how was the counting of GO Term Description performed? Was it just summing the counts of significantly enriched GO terms (again the cutoff of calling significance was nowhere to find) and representing them with a higher level parent GO term? The GO terms form a hierarchical tree and inherently some of the parent nodes will have more children nodes than others.  "RNA-binding was the most frequent term" might be simply due to the fact that the RNA-binding term has more children nodes than other terms shown in the figure. 
  7. In Fig.4B the total number of U1-lncRNAs associated mRNAs is ~3.9K, which does not match with the number (4350) reported in line 197. Some explanation is needed. Also, does the overlaps of 922 genes significant? A simple fisher exact test would be sufficed.
  8. In Fig.4E and lines 247-249, how was the "enrichment score of transcription cycle" defined? What were "a set of genes"? These should be clearly described in the methods.
  9. In lines 264-266, 45 "shared" U1-lncRNAs between the TCGA and GSE dataset were used as input in the LASSO model. However, in line 176 only 36 U1-lncRNAs were identified with the TCGA dataset. How can you get 45 shared(intersect) lncRNAs then? Was it the union set from the two datasets? It's unclear whether the GSE dataset was used in other analyses throughout the manuscript or only used in this prognostic model. Also noted the GSE76427 dataset is from microarray (which was never mentioned in the manuscript). It's known that the RNA expression values from microarray and RNA-seq follow different distributions. Perhaps this is the reason why the authors didn't achieve as good performances in the microarray dataset using the model defined with the TCGA dataset. This issue should be at least discussed in the Discussion section.

Minor issues:

  1. It's probably better to use the full name of TCGA-LIHC and SNRPD2 in the abstract when they were first introduced. Also in Fig.1A it says "TCGA-HCC". It would be better to keep it consistent across the manuscript and use TCGA-LIHC instead.
  2. In Methods 2.6 and 2.7, also mention cells transfected with NC.
  3. In line 160, "OS differences" should be overall survival differences?
  4. Fig.1D, no legends for the line length (said to be mapped with the correlation values) were provided.
  5. In line 209, "(U1-lncRNAs, hereafter)" is not needed as it was introduced in the previous section (lin 195).
  6. The texts in Fig.3 are hard to read.
  7. In Fig6, please state how many replicates were used in each of the cell experiments. Also, describe what tests were performed to measure the differences. Maybe also provide some quantification of the Western blot results.

Author Response

Response to Reviewer 1 Comments

Dear editor and dear reviewers

Re: Manuscript ID: diagnostics-1646872 and Title: U1RNP/lncRNA/transcription cycle axis promotes tumorigenesis of hepatocellular carcinoma

Thank you for your letter and the reviewers’ comments concerning our manuscript entitled “U1RNP/lncRNA/transcription cycle axis promotes tumorigenesis of hepatocellular carcinoma” (diagnostics-1646872). Those comments are very valuable and helpful. We have read through comments carefully and have made some corrections. Based on the instructions provided in your letter, we uploaded the file of the revised manuscript. Revisions in the text are shown using red highlight for additions. The responses to the reviewer’s comments are marked in light blue and presented following.

We would love to thank you for allowing us to resubmit a revised copy of the manuscript and we highly appreciate your time and consideration.

Major issues:

Q1: The concept of U1-lncRNAs is the key to all the analyses in the manuscript, but the authors did not give adequate descriptions of how they defined U1-lncRNAs.

Response:We are grateful for the suggestion. To be more clearly and in accordance with the reviewer’s concerns, we have revised more detailed interpretation about the concept of U1-lncRNAs in the manuscript.

Q1.1: In line 64, how exactly was the U1-binding motif computed? What are the "constitutive exons" used to compute the 9-nt motif? The HEXEvent is a database of exon splicing events published in 2013 and the website seems to be down now. I can only guess the authors used the sequences at splicing events and checked for motif enrichment. But without further details, there is no way to tell.

Response: We agree with the comment and added detailed information in the revised manuscript. The U1-binding motif was calculated by the following procedure:

We calculated the constitutive U1-binding site motif of human genome in the 5’ splice site with 3nt in exon and 6nt in intron, according to Ref [1]. We downloaded all annotated constitutive exons (exons without any known event of alternative splicing) of human genome version hg38 from HEXEvent database [2] (http://hexevent.mmg.uci.edu/cgi-bin/HEXEvent/HEXEventWEB.cgi). Sequence logo was plotted using the R package “ggplot2” and “ggseqlogo”.

We updated these details in the part 2.2. of methods.

Q1.2: In lines 66-68, what were the parameters used in FIMO? These will determine how stringent it was when calling the U1 motif is "in" the lncRNAs. Also, how did the authors define lncRNAs and how many lncRNA were used? Did they use annotations from GENCODE or did they manually curate the list (by length for example)?

Response:We agree with the comment and added detailed information in the revised manuscript. The criteria of p-value of FIMO scanning was set as less than 0.0001. A total of 18090 annotated lncRNAs from GENCODE were input and 2749 eligible lncRNAs were identified to contain U1-binding motif.

Q1.3. In lines 69-71, it seems that the Pearson's correlations were computed for each lncRNA-mRNA pair, across samples. How many tumor and normal samples in the TCGA dataset were used to compute the correlations? What are the standards to call the correlations to be significant? It seems they used p < 0.05 and |R|>0.5 (stated in main text line 177). Are those p-values adjusted for multiple comparisons (it seems to be raw p-values)? These are extremely important because the author said "If the correlation of lncRNA-mRNA pair was significant only in tumor but not in normal tissue, the lncRNA was identified as tumor specific lncRNA." Note that from Fig.2C it looks like there are much more tumor samples compared to normal samples. This will bias the correlation test where the expression of the to-test lncRNA-mRNA pair in the tumor samples might be easier to reach a significant correlation even the R values are relatively low. Also, it's unclear whether both tumor and normal samples or only tumor samples were used to compute Pearson correlation for the subsequent analyses, once the tumor-specific lncRNAs were defined.

Response:We are extremely grateful to reviewer for pointing out this problem. The full TCGA-LIHC dataset was used to compute the Pearson’s correlation efficient of each lncRNA-mRNA pair, which contained 371 tumor tissues and 50 normal tissues. The criteria for correlation were: p <0.05 and |R| >0.5. Indeed, the bigger sample size may result in higher chance of bias in raw p-value. To determine “tumor specific lncRNA-mRNA pairs”, we computed the correlation in tumor and normal tissues respectively. Subsequently, eligible lncRNA-mRNA pairs in tumor were selected while eligible ones in normal tissues were excluded. Thus, we obtained a list of tumor specific lncRNA-mRNA pairs. And the correlation of these pairs in only tumor tissues were used in the downstream analyses.

The following description was updated in the revised manuscript for detailed information about identification of U1-lncRNAs and correlated mRNAs:

The calculation procedure of U1-binding motif has been provided in Response to Q1.1 above. Subsequently, we downloaded all the long non-coding RNA transcript sequences (version 38) from GENCODE (https://www.gencodegenes.org/).We identified lncRNAs with U1-binding motif by FIMO scanning [3] (https://meme-suite.org/meme/tools/fimo). Then the sequence of all 18090 lncRNAs from the above step were input in FIMO and we obtained 2749 lncRNAs with match p-value less than 0.0001. 

We identified a total of 1027 eligible lncRNAs in TCGA dataset overlapped with the above list for subsequent analysis. Furthermore, we filtered lncRNAs and mRNAs according to the inclusion criteria: FPKM>1 in more than 90% of samples in the TCGA-LIHC expression matrix. Then a total of 1017 lncRNAs and 7565 mRNAs were eligible in tumor tissues, while a total of 962 lncRNAs and 7535 mRNAs were eligible in normal tissues.

Afterwards, we calculated the Pearson’s correlation of the above lncRNAs and mRNAs in the expression matrix data of tumor(N=371) and normal(N=50) tissues, respectively. To identify tumor specific lncRNA-mRNA pairs, we set the criteria: p<0.05; |R|>0.5. Eligible lncRNA-mRNA pairs in tumor tissues were included and then we excluded eligible lncRNA-mRNA pairs in normal tissues. Here, we identified tumor specific lncRNA-mRNA pairs, in which a total of 327 lncRNAs were identified as tumor specific lncRNAs.  

We further investigate the cellular location of the above lncRNAs with U1-binding motif in TCGA-LIHC dataset. Firstly, we examined the cellular location of all the 24514 lncRNAs from lncAtlas Database [4] (https://lncatlas.crg.eu) according to the mean Cytoplasmic/Nuclear value of all the cell lines tested. The number of nucleus-localised lncRNAs and cytoplasm- localised lncRNAs were 14342 and 10172, respectively. Then the intersect of lncRNAs with U1-binding motif and lncRNAs from lncAtlas was taken. A total of 509 lncRNAs with U1-binding motif could be found in lncAtlas, with 382 lncRNAs to be nucleus-localised and 127 lncRNAs to be cytoplasm- localized. A hypergeometric test was performed by using the function “HYPGEOMDIST” in Excel to compare the distribution of lncRNAs with U1-binding motif and all lncRNAs from lncAtlas. Finally, we identified a total of 175 nuclear U1-related tumor specific lncRNAs from the 327 tumor specific lncRNAs and then termed as U1-lncRNAs for subsequent analysis.

Q2: In Fig.1B, how was the differential expressed tested? Did the authors run a standard RNA-seq DE test (like edgeR, DEseq2, etc.) or it was just a simple t-test or Wilcoxon test with the FPKM? How was "significantly higher" defined (in line 173)? What are the meaning of those asterisks on top (FDR<0.001?) These should be stated clearly in the methods and the figure legend.

Response: We deeply appreciate the reviewer’s suggestion. For fig.1B, we updated the related methods and figure legend. The expression of U1RNPs in each sample was normalized by log2(FPKM+1) and T-test was used for comparison. Significance level: **** p< 0.0001.

Q3: In Fig.2A and line 189, the authors said the U1-lncRNAs are predominantly located in the nuclear. What are the distributions of the nuclear/cytoplasm localization for other lncRNAs in the LncAtlas database? Without some sort of proper background to compare with, it's hard to see whether the preference in the nuclear is a property of the U1-lncRNAs or it's universal for all lncRNAs.

Response: Thank you for the suggestion. As is mentioned in the response to Q2, a hypergeometric test was performed to compare the distribution of lncRNAs with U1-binding motif and all lncRNAs from lncAtlas. The statistical p-value of hypergeometric test was less than 0.0001, which means significant higher nucleus-localized ratio of the lncRNAs with U1 binding motif.

Q4: In the figure legend of Fig.2B, what does "according to their expression level" mean? It seems that the counts in the bar plots are the numbers of lncRNAs associated with each of those U1RNPs. No gene expression levels (either the lncRNAs or the U1RNPs) were shown here.

Response: Thank you for the suggestion. The figure legend of fig.2B in the previous version was not accurate. We revised the legend: Bar plot showing the overall significant counts and location of correlated lncRNAs of each U1RNP based on the correlation of their expression level. The criteria of significant correlation were: p<0.05; |R| >0.5.

Q5: What were the background genes to test against in the GO enrichment analysis?

Response: Thank you for the suggestion. The GO enrichment analysis was performed by the R package “ClusterProfiler” and the “GO-BP, MF and CC” ontology was used as background. We revised these content in Methods.

Q6: In Fig.4A and lines 233-234, how was the counting of GO Term Description performed? Was it just summing the counts of significantly enriched GO terms (again the cutoff of calling significance was nowhere to find) and representing them with a higher level parent GO term? The GO terms form a hierarchical tree and inherently some of the parent nodes will have more children nodes than others.  "RNA-binding was the most frequent term" might be simply due to the fact that the RNA-binding term has more children nodes than other terms shown in the figure. 

Response: We are grateful for the suggestion. The GO level count was performed by TBtools[5] based on data downloaded from http://geneontology.org/docs/download-ontology/, and only Level 5 GO terms in GO-molecular function (MF) ontology category were counted. The MF category contains “RNA-binding” and was selected. For the concern of too many children nodes of a certain parent GO term, we visited Gene Ontology (GO) knowledgebase (http://amigo.geneontology.org/) and checked some terms in our count result. For instance, kinase activity (GO:0016301) has more children nodes than RNA-binding (GO:0003723), while the count of the former one was 123 and the latter was 234 in our result. There is no denying that GO term count was not an accurate method due to the complexity of the database, whereas the purpose of this analysis was merely to explore the association of RNA-binding genes with U1-lncRNAs.

Q7: In Fig.4B the total number of U1-lncRNAs associated mRNAs is ~3.9K, which does not match with the number (4350) reported in line 197. Some explanation is needed. Also, does the overlaps of 922 genes significant? A simple fisher exact test would be sufficed.

Response: Thank you for the suggestion. We checked the unmatched number and found it was a mistake that hasn’t been fixed in previous versions of analysis due to our carelessness. The number was revised as follows:

Filtering with the Pearson’s correlation more than 0.5 or less than -0.5, a total of 3453 mRNAs were significantly correlated with U1-lncRNAs (3252 mRNAs were positive correlated and 201 mRNAs were negative correlated).

What’s more, the Venn result was corrected as well, which contained 851 overlap genes. In addition, to examine the significance of the overlap, a fisher’s exact test was used and the background gene number was set as 19969 [6].

Q8: In Fig.4E and lines 247-249, how was the "enrichment score of transcription cycle" defined? What were "a set of genes"? These should be clearly described in the methods.

Response: Thank you for the suggestion. The list of selected genes in transcription cycle pathway were from a reference literature [7], which contains a total of 108 related genes (A detailed list can be found in Supplementary Table 1). We revised the above detail in Methods.

Q9: In lines 264-266, 45 "shared" U1-lncRNAs between the TCGA and GSE dataset were used as input in the LASSO model. However, in line 176 only 36 U1-lncRNAs were identified with the TCGA dataset. How can you get 45 shared(intersect) lncRNAs then? Was it the union set from the two datasets? It's unclear whether the GSE dataset was used in other analyses throughout the manuscript or only used in this prognostic model. Also noted the GSE76427 dataset is from microarray (which was never mentioned in the manuscript). It's known that the RNA expression values from microarray and RNA-seq follow different distributions. Perhaps this is the reason why the authors didn't achieve as good performances in the microarray dataset using the model defined with the TCGA dataset. This issue should be at least discussed in the Discussion section.

Response: We are extremely grateful to reviewer for pointing out this problem. We identified a total of 175 U1-lncRNAs. The number of shared lncRNAs was 45 in the intersect union of both TCGA-LIHC and GSE76427 dataset. In addition, the latter dataset was only used for the construction of prognostic model, which was stated in the revised manuscript.

However, to achieve better performance of the model, we reconstructed the model based on both U1-lncRNAs and their associated mRNAs. The result was revised and the model was better in AUC analysis. The U1-LncRNA risk model performed well in the TCGA-LIHC cohort to predict overall survival with the 1-,3- and 5- AUC to be 0.82, 0.84 and 0.8, respectively. And the 1-,3- and 5- AUC of the U1-lncRNA risk model in GSE76427 cohort were 0.71, 0.71 and 0.62, respectively.

Unfortunately, the p-value of KM curves was 0.2, which didn’t reach the significant criteria. As the review mentioned, the possible reason maybe the different methods of measurement and normalization. We also discussed this point in the revised Discussion section.

Minor issues:

Q1: It's probably better to use the full name of TCGA-LIHC and SNRPD2 in the abstract when they were first introduced. Also in Fig.1A it says "TCGA-HCC". It would be better to keep it consistent across the manuscript and use TCGA-LIHC instead.

Response: We are grateful for the suggestion. The full name of TCGA-LIHC and SNRPD2 were revised in the abstract. And the caption in Fig.1A was also revised from “TCGA-HCC” to “TCGA-LIHC”.

Q2: In Methods 2.6 and 2.7, also mention cells transfected with NC.

Response: We are grateful for the suggestion. The cells transfected with siNC were also mentioned in Methods 2.6 and 2.7.

Q3: In line 160, "OS differences" should be overall survival differences?

Response: We are grateful for the suggestion. The OS represented overall survival and “OS differences” was revised as “overall survival (OS) differences”.

Q4: Fig.1D, no legends for the line length (said to be mapped with the correlation values) were provided.

Response: We are grateful for the suggestion. We didn’t map line length with any data. Actually, for better visualization and layout of the graph, we adjusted manually the location of some pies in CytoScape, which may change the original line. Moreover, we found the information about size of pies were missed and revised as “mapping with the expression level”.

Q5: In line 209, "(U1-lncRNAs, hereafter)" is not needed as it was introduced in the previous section (lin 195).

Response: We are grateful for the suggestion. The redundant “(U1-lncRNAs, hereafter)” in line 209 was removed.

Q6: The texts in Fig.3 are hard to read.

Response: We are grateful for the suggestion. We cut the Fig.3 into a new Fig.3 (containing previous A and B) and a new Fig.S1(containing previous C, D and E). The order of previous Fig.S1 and S2 was changed correspondingly to Fig.S2 and S3.

Q7: In Fig.6, please state how many replicates were used in each of the cell experiments. Also, describe what tests were performed to measure the differences. Maybe also provide some quantification of the Western blot results.

Response: We are grateful for the suggestion. All cell experiments were performed for three biological replicates. A grey scale quantification of Western Blot was performed by ImageJ. The statistical method used in each test was added in correspond part. These points were revised in our manuscript.

Reference:

  1. Sheth N, Roca X, Hastings M, Roeder T, Krainer A, Sachidanandam R. Comprehensive splice-site analysis using comparative ge-nomics. Nucleic acids research. 2006;34(14):3955-67.
  2. Busch A, Hertel KJ. HEXEvent: a database of Human EXon splicing Events. Nucleic acids research. 2013;41(Database issue):D118-24.
  3. Grant CE, Bailey TL, Noble WS. FIMO: scanning for occurrences of a given motif. Bioinformatics. 2011;27(7):1017-8.
  4. Mas-Ponte D, Carlevaro-Fita J, Palumbo E, Hermoso Pulido T, Guigo R, Johnson R. LncATLAS database for subcellular localiza-tion of long noncoding RNAs. RNA (New York, NY). 2017;23(7):1080-7.
  5. Chen C, Chen H, Zhang Y, Thomas HR, Frank MH, He Y, et al. TBtools: An Integrative Toolkit Developed for Interactive Analyses of Big Biological Data. Molecular plant. 2020;13(8):1194-202.
  6. Nurk S, Koren S, Rhie A, et al. The complete sequence of a human genome. Science. 2022;376(6588):44-53.
  7. Vervoort SJ, Devlin JR, Kwiatkowski N, Teng M, Gray NS, Johnstone RW. Targeting transcription cycles in cancer. Nat Rev Cancer. 2022;22(1):5-24.

Reviewer 2 Report

A well written manuscript

Author Response

We appreciate the reviewer’s positive evaluation of our work.

Round 2

Reviewer 1 Report

The authors have addressed all my previous comments pretty well and I think the manuscript is now in a good shape.